# Characteristics of Vaccine- and Infection-Induced Systemic IgA Anti-SARS-CoV-2 Spike Responses

**DOI:** 10.3390/vaccines11091462

**Published:** 2023-09-07

**Authors:** Natasha J. Norton, Danielle P. Ings, Kathleen E. Fifield, David A. Barnes, Keeley A. Barnable, Debbie O. A. Harnum, Kayla A. Holder, Rodney S. Russell, Michael D. Grant

**Affiliations:** 1Immunology & Infectious Diseases Program, Division of BioMedical Sciences, Faculty of Medicine, Memorial University of Newfoundland, St. John’s, NL A1B 3V6, Canada; njnorton@mun.ca (N.J.N.); dings@mun.ca (D.P.I.); c37kef@mun.ca (K.E.F.); dab866@mun.ca (D.A.B.); keeleyb@mun.ca (K.A.B.); kayla.holder@mun.ca (K.A.H.); rodney.russell@med.mun.ca (R.S.R.); 2Newfoundland and Labrador Health Services, St. John’s, NL A1B 3V6, Canada; debbie.harnum@easternhealth.ca

**Keywords:** SARS-CoV-2, IgA, spike, breakthrough infection, Omicron, vaccination, imprinting

## Abstract

Mucosal IgA is widely accepted as providing protection against respiratory infections, but stimulation of mucosal immunity, collection of mucosal samples and measurement of mucosal IgA can be problematic. The relationship between mucosal and circulating IgA responses is unclear, however, whole blood is readily collected and circulating antigen-specific IgA easily measured. We measured circulating IgA against SARS-CoV-2 spike (S) to investigate vaccine- and infection-induced production and correlation with protection. Circulating IgA against ancestral (Wuhan-Hu-1) and Omicron (BA.1) S proteins was measured at different time points in a total of 143 subjects with varied backgrounds of vaccination and infection. Intramuscular vaccination induced circulating anti-SARS-CoV-2 S IgA. Subjects with higher levels of vaccine-induced IgA against SARS-CoV-2 S (*p* = 0.0333) or receptor binding domain (RBD) (*p* = 0.0266) were less likely to experience an Omicron breakthrough infection. The same associations did not hold for circulating IgG anti-SARS-CoV-2 S levels. Breakthrough infection following two vaccinations generated stronger IgA anti-SARS-CoV-2 S responses (*p* = 0.0002) than third vaccinations but did not selectively increase circulating IgA against Omicron over ancestral S, indicating immune imprinting of circulating IgA responses. Circulating IgA against SARS-CoV-2 S following breakthrough infection remained higher than vaccine-induced levels for over 150 days. In conclusion, intramuscular mRNA vaccination induces circulating IgA against SARS-CoV-2 S, and higher levels are associated with protection from breakthrough infection. Vaccination with ancestral S enacts imprinting within circulating IgA responses that become apparent after breakthrough infection with Omicron. Breakthrough infection generates stronger and more durable circulating IgA responses against SARS-CoV-2 S than vaccination alone.

## 1. Introduction

The COVID-19 pandemic inspired massive scientific and clinical research efforts that introduced and distributed vaccines against SARS-CoV-2 within a uniquely accelerated time frame. As of July 2023, more than 13 billion vaccine doses have been administered globally, over 98 million of which were administered within Canada [1]. While these vaccines continue to protect against severe illness, they fail to provide sterilizing immunity against emerging SARS-CoV-2 variants [2], as clearly illustrated by the widespread occurrence of Omicron breakthrough infections. Ongoing diversification of SARS-CoV-2 raised concerns around the ability of ancestral Wuhan-Hu-1-based SARS-CoV-2 vaccines to continue providing protection against illness, prompting the introduction of bivalent mRNA vaccines encoding both ancestral and Omicron spike (S) antigens. 

While IgG subclass antibodies (Ab) reach the highest levels in circulation, IgA Ab dominate at mucosal sites, where they may play a more significant role in protection from respiratory and other mucosal infections. Unfortunately, systemically administered, non-replicating vaccines are conspicuously poor at inducing mucosal IgA responses. Following initial antigenic exposure, responding B cells secrete immunoglobulin (Ig) M subclass Ab with relatively low affinity and poor tissue penetration. With T cell help, proliferating B cells in newly formed germinal centres undergo somatic hypermutation and isotype switching, differentiate into plasmablasts and potentially become plasma cells. Isotype switching to IgG predominates systemically, while isotype switching to IgA predominates mucosally. Most IgA produced mucosally is secreted as a dimer into mucosal fluids lining the oronasal, esophageal, lower respiratory, gastrointestinal and urogenital tracts, which are sites of entry and/or replication for many pathogens. Monomeric IgA circulates in the bloodstream and there are conflicting reports on the relationship between mucosal and systemic IgA responses [3,4]. Mucosal IgG composition reflects spillover from the circulation, but commonalities in origin between systemic and mucosal IgA remain undefined [5,6]. 

While it is mucosal IgA that is most likely to contribute protection from respiratory infections, for ease and simplicity, vaccine-induced Ab production studies primarily focus on the capacity of circulating IgG to neutralize SARS-CoV-2 variants [7,8,9,10,11]. Vaccine-induced anti-SARS-CoV-2 IgG responses are optimized with sufficient time intervals between doses, yet fall short of responses seen with hybrid immunity [12,13,14,15]. Furthermore, subjects who experience an infection with SARS-CoV-2 variants following vaccination with ancestral-S-based vaccines continue to display preferential IgG responses against ancestral S. This favouring of an immune response toward previously encountered, closely related versions of an extant antigen is known as original antigenic sin or immune imprinting [16]. Such a preference to reactivate existing memory B cells at the expense of de novo B cell activation potentially reduces the ability to neutralize emerging variants [16]. While imprinting is well described for circulating IgG, there has been less study of this phenomenon with circulating or mucosal IgA. 

In this study, we investigated circulating IgA anti-SARS-CoV-2 S responses after vaccination and after a breakthrough infection (SARS-CoV-2 infection after receiving at least two COVID-19 vaccines) to better understand the impact of an Omicron breakthrough infection relative to vaccination on anti-SARS-CoV-2 S IgA levels. As intramuscular vaccination delivers antigens differently than infection and does not favour IgA production, we also assessed whether vaccination with an ancestral S caused imprinting of the IgA response in the context of Omicron breakthrough infection. By comparing circulating IgA anti-SARS-CoV-2 S levels in subjects who became infected after vaccination to levels in those who remained uninfected, we tested whether circulating IgA anti-SARS-CoV-2 S or IgA anti-SARS-CoV-2 S receptor binding domain (RBD) levels can predict protection from infection.

## 2. Materials and Methods

### 2.1. Selection of Subjects and Blood Sample Processing

This research was approved by the Health Research Ethics Board of Newfoundland and Labrador. The investigation was conducted following guidelines outlined in the Canadian Tri-Council Policy Statement: Ethical Conduct for Research Involving Humans. The study’s participants are part of an ongoing research cohort at Memorial University of Newfoundland and Labrador, and all subjects within this research study were selected based on prior suspicion of SARS-CoV-2 infection or confirmation with reverse-transcriptase polymerase chain reaction (RT-PCR) [17]. In accordance with the Declaration of Helsinki, written informed consent was obtained for the collection of whole blood. As part of the study intake process, participants completed a questionnaire regarding their history of SARS-CoV-2 exposure, testing, and symptoms experienced. Individuals who self-reported Omicron infections—between February and August 2022—based on RT-PCR or rapid test results, following receipt of at least two Health-Canada-approved anti-SARS-CoV-2 vaccines, were selected for further study. Infection was also confirmed serologically via detection of antibodies selective for SARS-CoV-2 nucleocapsid (N) [17]. We measured anti-SARS-CoV-2 N antibody levels longitudinally to determine whether and when subjects were exposed to the virus (either the original Wuhan-Hu-1 strain or Omicron) as opposed to being vaccinated with a vaccine in which only the S protein of SARS-CoV-2 was represented. This assay was used to confirm that there was no infection prior to vaccination and then to confirm breakthrough infection following vaccination in those reporting PCR or rapid antigen tests documenting infection. Whole blood was collected through forearm venipuncture and drawn into acid–citrate–dextrose vaccutainers to prevent clotting. Following a 10 min centrifugation at 500× *g*, plasma was collected and promptly stored at −80 °C until testing. 

### 2.2. Assessment of Circulating IgG and IgA Anti-SARS-CoV-2 S Levels Using ELISA

Plasma was thawed on ice and diluted with phosphate-buffered saline (PBS) containing 0.05% Tween 20 and 0.1% bovine serum albumin (BSA, Sigma-Aldrich, St. Louis, MO, USA). Anti-SARS-CoV-2 Ab levels were measured against recombinant proteins with an enzyme-linked immunosorbent assay (ELISA) using 96-well Immulon-2 HB (Thermo Fisher Scientific, Rochester, NY, USA) ELISA plates. Overnight coating of antigens with 50 ng protein/well in 50 μL of Dulbecco’s PBS (Corning, Mediatech, Inc., Manassa VA, USA) was conducted at 4 °C. Anti-SARS-CoV-2 Ab activity was measured using ELISA against Wuhan-Hu-1 full-length spike (FLS, SMT1-1 Wuhan-Hu-1, National Research Council of Canada), as well as RBD (Sino Biological, Wayne, PA, USA) and SARS-CoV-2 Omicron BA.1 FLS (SMT1-1 Omicron BA.1, National Research Council of Canada) and RBD (Omicron BA.1, ACROBiosystems, Newark, DE, USA). After overnight coating, plates underwent four washes, followed by six washes between subsequent steps, with 300 μL/well of PBS containing 0.05% Tween 20. Plates were then blocked with 200 μL of 1% BSA in PBS for one hour. Following this, 100 μL of diluted plasma was added and left for 1.5 h followed by a 1h incubation with 100 μL/well of diluted goat-anti human IgG or IgA horseradish peroxidase (HRP)-conjugated antibodies (Jackson ImmunoResearch Baltimore Pike, West Grove, PA, USA). For colour development, 100 μL of 3,3’,5,5’-tetramethylbenzidine (TMB, OptEIA™ Substrate Reagents A & B, BD Biosciences, San Diego, CA, USA) was added for 20 min, followed by 100 μL of 1 M H_2_SO_4_ (Sigma-Aldrich) to stop the reaction. Optical density (OD) was measured at 450 nm using a BioTek Synergy HT plate reader. Working plasma dilutions for assessing IgG and IgA antibody levels were 1:100 and 1:50, respectively, and conjugate dilutions were 1:50,000 for anti-IgG*HRP and 1:25,000 for anti-IgA*HRP. Variability and reproducibility of this assay format have been described [12,15,17]. All between-group comparisons shown were run in the same assay. 

### 2.3. Statistical Analysis

GraphPad Prism Version 9.5.1 was used for all statistical analyses. When indicated, the probability of significance difference values were presented above lines extending across the groups compared. As not all data were normally distributed, non-parametric Mann–Whitney, Wilcoxon signed rank, and Spearman correlation tests were used to compare medians between groups, assess differences in paired data and assess correlation, respectively.

## 3. Results

### 3.1. Study Cohort

For an ongoing study of immunity against SARS-CoV-2 initiated in March 2020, individuals in Newfoundland and Labrador were recruited based on confirmed or suspected SARS-CoV-2 infection. The subjects provided whole blood samples every 3 months throughout their course of vaccinations against COVID-19 and potential exposures to SARS-CoV-2. We selected non-immunocompromised individuals with documented evidence of infection following at least two doses of Canadian-Public-Health-Agency-approved vaccines, Pfizer BioNTech (BNT162b2), Moderna (mRNA-1273), and AstraZeneca (ChAdOx1). These individuals self-reported a positive SARS-CoV-2 antigen rapid test or RT-PCR result between 21 February and 8 August 2022. All had increases in IgG anti-SARS-CoV-2 N protein levels indicative of infection [17], and in most cases, experienced symptoms of COVID-19. Humoral immune responses of this group of subjects identified as having breakthrough infections were compared to groups of subjects with at least two COVID-19 vaccinations that did not experience breakthrough infection. The first confirmed case of the SARS-CoV-2 Omicron variant was detected in Canada on 29 November 2021 [18] and in Newfoundland and Labrador (NL) on 15 December 2021 [19]. Omicron quickly became the dominant strain in NL and across Canada. Although our subjects’ infections were not typed for the SARS-CoV-2 variant, based on the time since the first case in NL, we assumed the majority of infections occurring over the study period were with a SARS-CoV-2 Omicron variant or subvariant (Table 1). We further separated subjects into comparison groups based on the number of COVID-19 vaccinations received. Group size, subject sex, and age, together with information on vaccine types and the number of days after vaccination or infection that samples were collected are summarized in Table 1.

### 3.2. Intramuscular Vaccination Induces Circulating IgA 

Plasma samples from all subjects were tested using ELISA for IgA anti-SARS-CoV-2 S and RBD responses after intramuscular vaccination. Our data confirm that intramuscular vaccination alone generates readily detectable circulating IgA anti-SARS-CoV-2 Wuhan-Hu-1 FLS (Figure 1a). A longitudinal assessment of plasma samples from 22 subjects indicated a significant increase in vaccine-induced circulating IgA responses following third vaccine doses (median OD IgA anti-SARS-CoV-2 FLS ± interquartile range (IQR) 0.36, 0.21–0.55 vs. 0.57, 0.30–0.83, *p* = 0.0006), however, IgA anti-SARS-CoV-2 S levels in plasma were significantly lower (0.57, 0.30–0.83 vs. 0.39, 0.30–0.65, *p* = 0.0053) after a fourth vaccination at time of measurement (Figure 1b). 

### 3.3. Plasma IgA Anti-SARS-CoV-2 S Levels and Breakthrough Infection

We compared levels of IgA anti-SARS-CoV-2 FLS and RBD in 57 subjects who experienced breakthrough infections to levels in 86 subjects who remained uninfected. The mean duration between the second and third vaccines is 192 days and between the third and fourth vaccines is 238 days. Subject blood samples were collected an average of 72, 44 and 58 days post vaccines 2, 3 and 4, respectively. Following two vaccinations, there was no significant difference in the amount of plasma IgA (Figure 2a,b) or IgG (Figure 2c) anti-SARS-CoV-2 S between groups (Figure 2c). The paucity of subjects remaining who had received only two vaccines limited the strength of this comparison. The subjects with two vaccines were roughly ten years younger than those receiving three vaccines and almost twenty years younger than those receiving four (Table 1). Following three vaccinations, the group who remained uninfected (n = 49) had stronger vaccine-induced plasma IgA responses against FLS (median OD ± IQR 0.50, 0.31–0.76 vs. 0.28, 0.17–0.61, *p* = 0.0471) and RBD (0.24, 0.13–0.46 vs. 0.13, 0.05–0.46, *p* = 0.0266) compared to subjects (n = 47) who later experienced breakthrough infections (Figure 2a,b). There was no significant difference in vaccine-induced anti-SARS-CoV-2 FLS IgG between the same groups (Figure 2c). In subjects who experienced breakthrough infections after three vaccines, there was no significant correlation between either vaccine-induced anti-FLS (Figure 2d) or anti-RBD (Figure 2e) IgA levels and days between vaccination and infection.

### 3.4. Effect of Omicron Breakthrough Infection on Plasma IgA

Within this study, the term immunogenic exposure refers to the total number of times a subject was exposed to SARS-CoV-2 antigens through either vaccination or infection. We compared IgA anti-SARS-CoV-2 FLS responses between subjects with the same number of immunogenic exposures, with and without SARS-CoV-2 infection. Breakthrough infection after two vaccines produced more robust circulating IgA responses than a third vaccine dose (median OD ± IQR, 1.24, 1.11–1.41 vs. 0.63, 0.34–1.06, *p* = 0.0003, Figure 3a). Breakthrough infection after three vaccines resulted in stronger circulating IgA responses than a fourth vaccine dose (median OD ± IQR, 1.20, 0.89–1.71 vs. 0.39, 0.26–0.61, *p* < 0.0001, Figure 3a). Longitudinal assessment of circulating IgA levels against SARS-CoV-2 Wuhan-Hu-1 (Figure 3b) and Omicron BA.1 FLS (Figure 3c) after two vaccines and a breakthrough infection and after three vaccines and an Omicron breakthrough infection (Figure 3d,e) showed that post-infection levels remained elevated over 150 days post infection.

### 3.5. Selective Recognition of Wuhan-Hu-1 S Is Preserved despite Omicron Breakthrough Infection

To investigate whether imprinting of systemic humoral responses against SARS-CoV-2 affects IgA responses similarly to IgG responses, we compared response levels against ancestral Wuhan-Hu-1 FLS versus Omicron BA.1 FLS after vaccination and again following an Omicron breakthrough infection. After receiving vaccines based on an ancestral Wuhan-Hu-1 S, persons vaccinated three (median OD ± IQR, 0.59, 0.31–1.03 vs. 0.63, 0.17–0.70, *p* < 0.0001) or four times (0.39, 0.26–0.61 vs. 0.23, 0.14–0.49, *p* < 0.0001) had significantly stronger anti-FLS IgA responses against Wuhan-Hu-1 than against Omicron BA.1 (Figure 4a), as expected. However, circulating IgA responses remained stronger against Wuhan-Hu-1 FLS even after Omicron breakthrough infection following two (median OD ± IQR, 1.22, 1.10–1.43 vs. 0.98, 0.87–1.25, *p* = 0.0010) or three (0.87, 0.61–1.42 vs. 0.67, 0.48–1.23, *p* = 0.0003) vaccinations (Figure 4a). Favoured recognition of ancestral Wuhan-Hu-1 FLS over Omicron BA.1 FLS was confirmed for circulating IgG after three vaccines (median OD ± IQR, 2.03, 1.78–2.44 vs. 1.77, 1.69–1.85, *p* = 0.0038), four vaccines (2.06, 2.03–2.36 vs. 1.51, 1.43–1.64, *p* < 0.0001), two vaccines followed by Omicron infection (2.21, 1.89–2.42 vs. 1.76, 1.72–1.83, *p* = 0.0005) and three vaccines (2.33, 2.04–2.46 vs. 1.62, 1.54–1.67, *p* < 0.0001) followed by Omicron infection (Figure 4b). Thus, immune imprinting from vaccination impacts SARS-CoV-2 FLS IgA and IgG responses.

## 4. Discussion

In this study, we investigated the impact of multiple vaccinations and breakthrough infection on SARS-CoV-2 S-specific circulating IgA. We confirmed that two intramuscular vaccinations with mRNA-based vaccines induce systemic IgA anti-FLS and anti-RBD responses against SARS-CoV-2 and that a third immunization boosts this response. However, the increase in circulating anti-SARS-CoV-2 S IgA from a third vaccination was lesser in both magnitude and durability compared to the increase observed from a breakthrough infection following two vaccinations. Persons experiencing breakthrough infections after two or three vaccines had higher levels of circulating IgA anti-SARS-CoV-2 S than persons receiving third or fourth vaccines. Fourth vaccinations had no significant effect on SARS-CoV-2 S-specific circulating IgA. Higher levels of vaccine-induced circulating anti-SARS-CoV-2 S IgA after three vaccinations were associated with a reduced risk of breakthrough infection, despite no similar association with vaccine-induced anti-SARS-CoV-2 S IgG levels. As with circulating IgG responses, vaccination with an ancestral SARS-CoV-2 S antigen imposed immunological imprinting on IgA responses with preferred recognition of ancestral SARS-CoV-2 S protein over Omicron SARS-CoV-2 S protein persisting following Omicron breakthrough infection. 

Intramuscular vaccines, like the mRNA-based vaccines approved for use against SARS-CoV-2, are taken up by antigen-presenting cells such as dendritic cells in the muscle and trafficked to the draining lymph node for presentation to T cells. Since the mRNA encoding SARS-CoV-2 S protein is translated inside host cells and the newly synthesized S protein is released for uptake by other cells, antigen presentation through MHC class I and II molecules occurs to prime both CD4^+^ and CD8^+^ T cells. Binding antigenic epitopes of SARS-CoV-2 S protein and interacting with CD4^+^ follicular helper T cells within lymph node germinal centres encourages B cells to undergo proliferation, somatic hypermutation and isotype class switching as they differentiate into Ig-secreting plasma cells or memory B cells. With boosting, the mRNA vaccines generate robust circulating anti-SARS-CoV-2 S Ab responses, where IgG reaches the highest levels and is most relevant for virus neutralization [14,20,21,22,23,24]. Circulating IgA against SARS-CoV-2 S also rises with booster vaccination and can contribute to neutralization, but IgA anti-SARS-CoV-2 S plateaus at lower levels than IgG. Although not administered in a manner to effectively induce mucosal immunity, intramuscular mRNA vaccines do generate local salivary responses, where IgA levels dominate over IgG [5,25]. The mucosal anti-SARS-CoV-2 S response can potentially block viral binding to host cells in the oral cavity, nasal and upper respiratory tract, thereby acting directly at sites of exposure to prevent infection. Induction of both serum and mucosal IgA responses after vaccination has also been described with oral rotavirus live attenuated vaccines and intramuscular replication-defective lentivirus vaccines [26,27].

While there is general agreement that mucosal IgA provides protection against respiratory infections, there is considerable controversy around the relevance of circulating antiviral IgA. Since IgA is present at easily measurable levels in plasma and serum, it would be valuable to know how levels and specificity relate to mucosal IgA activity in terms of protection from infection and protection from severe illness. Several studies reported that systemic IgA anti-SARS-CoV-2 is often detectable earlier than either IgM or IgG and that higher systemic levels of SARS-CoV-2-specific IgA detectable early in the course of infection are associated with severe illness [28,29,30]. Conversely, other studies found early systemic IgA levels were unrelated to disease severity or even associated with asymptomatic infections [24,31]. Discrepant results and interpretations of the significance of anti-SARS-CoV-2 IgA in these studies likely reflect variation in the study designs, subject populations (hospitalized versus community) and timing of Ab measurements (acute or convalescent). Our study was not designed or powered to compare the immunogenicity of different mRNA vaccines. 

The earlier appearance of IgA anti-SARS-CoV-2 at higher levels than IgM or IgG systemically early after infection would not be predicted, based on the time required for transition from a primary to secondary response and the greater frequency of isotype switching to IgG compared to IgA. One possible explanation is that the early-phase systemic IgA response in these cases reflects the production of natural antibodies from activation of B1 B cells expressing IgA [32]. The activation of B1 B cells was previously associated with failure to clear acute hepatitis C virus infection, so it may signify failure to rapidly activate potent antiviral adaptive immunity [33]. In contrast, higher IgA levels during convalescence might signify the development of robust humoral immunity, enhanced viral clearance and some level of protection against future infection. Both our results and those of Sheikh-Mohamed et al., 2022 [5] indicate that higher levels of vaccine-induced circulating IgA, but not IgG anti-SARS-CoV-2 S Ab, are associated with some aspect of protection from SARS-CoV-2 infection. 

As there are conflicting reports regarding the correlation between circulating and mucosal IgA Ab levels, the relationship between circulating and mucosal IgA responses is uncertain [3,4,34,35,36]. Although circulating IgA is not itself protective at the site of infection, if levels and specificity of mucosal and circulating IgA are closely related, it could serve as an easily measured marker for mucosal antiviral IgA activity. Robust mucosal and systemic responses might be induced independently, or there could be a mutual spillover of IgG and IgA between mucosal and systemic sites. Comparison of variable (V) region sequences between circulating IgG and IgA anti-SARS-CoV-2 S and between mucosal and circulating anti-SARS-CoV-2 S-specific B cells is required to address the origin of circulating SARS-CoV-2 S-specific IgA and the nature of its relationship to mucosal IgA. The association between plasma IgA anti-SARS-CoV-2 S levels and protection from infection that we and others observed [5], which was not found for IgG anti-SARS-CoV-2 S levels, suggests that circulating IgA is not simply a fractional representation of the IgG responses reflecting the lower frequency of systemic isotype switching to IgA. 

Despite breakthrough infection after two vaccinations having a greater impact on circulating anti-SARS-CoV-2 IgA levels and durability than a third vaccination, we observed vaccine-induced imprinting of the IgA response to SARS-CoV-2 S antigen. Following infection with Omicron, Ab reactivity against the ancestral Wuhan-Hu-1 S protein represented in the vaccine remained higher than reactivity against the S protein of the infecting variant. Despite remaining lower than reactivity against vaccine-encoded S in all but a few cases, IgA reactivity against Omicron S was increased by infection, illustrating the value of boosting humoral responses to higher levels, even with imprinting. A similar level of imprinting for IgG and IgA anti-SARS-CoV-2 S Ab suggests significant overlap in B cell clones selected through mRNA vaccination for affinity maturation and isotype switching to IgG and IgA. Whether imprinting of IgA responses through repeated exposure to ancestral antigens occurs similarly to that of IgG responses through variable isotype switching of overlapping B cell clones or through a separate pathway of B cell activation and maturation remains to be demonstrated via V region analysis. Likewise, whether spillover of IgA between systemic and mucosal sites occurs requires further analysis. Given that SARS-CoV-2 enters through the respiratory tract and that the respiratory tract is at least initially its primary site of replication, the impact of infection on mucosal IgA anti-SARS-CoV-2 S levels would likely be much greater than that of an intramuscular vaccine. The greater impact on circulating IgA anti-SARS-CoV-2 S may also reflect greater antigen exposure and/or spillover from IgA induced at mucosal sites; however, no saliva or other mucosal fluids were collected for this study to test this possibility. 

While more information is required on the relationship between mucosal and circulating IgA to clearly establish the relevance of circulating IgA in protection against respiratory viruses such as SARS-CoV-2, several factors illustrate value in its measurement. Intramuscular mRNA vaccination induces a circulating IgA response that generally parallels the IgG response in specificity at lower but easily measured levels. Our study corroborated a previously reported association between higher levels of circulating anti-SARS-CoV-2 S IgA and protection from a breakthrough infection, illustrating a direct or indirect relationship to protective immune features [5]. The relevance that circulating IgA responses have toward protection from infection and/or toward limiting the duration and severity of infection warrants further study.

## 5. Conclusions

This study examined the effects of vaccination and breakthrough infection on circulating IgA responses against SARS-CoV-2 S. Intramuscular mRNA vaccines induced systemic IgA responses that were boosted by third vaccinations and higher IgA anti-SARS-CoV-2 S levels were associated with protection from breakthrough infection. However, a breakthrough infection caused a more robust increase in IgA responses compared to vaccination. Immunological imprinting of IgA responses occurred from vaccines encoding ancestral S in that recognition of ancestral SARS-CoV-2 S protein was favoured over recognition of the Omicron S protein even after an Omicron breakthrough infection. Elucidating a relationship between circulating and mucosal anti-SARS-CoV-2 IgA will be important to further document value in measuring circulating vaccine- and infection-induced IgA responses. 

## Figures and Tables

**Figure 1 vaccines-11-01462-f001:**
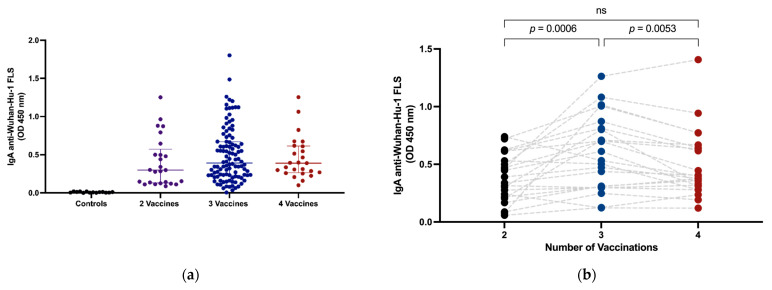
Circulating intramuscular vaccine-induced SARS-CoV-2 S-specific IgA. Plasma IgA OD levels against SARS-CoV-2 Wuhan-Hu-1 FLS were measured in pre-pandemic controls and study subjects grouped by number of vaccines received (**a**). Plasma IgA levels against SARS-CoV-2 Wuhan-Hu-1 FLS were measured longitudinally in subjects following second, third and fourth vaccines (**b**). Horizontal lines within the grouped data points (**a**) represent median plus or minus interquartile range (IQR). Comparison of median values between groups was performed using the Wilcoxon signed rank test (**b**). When significant, probability values for differences between groups are shown above lines spanning the groups compared.

**Figure 2 vaccines-11-01462-f002:**
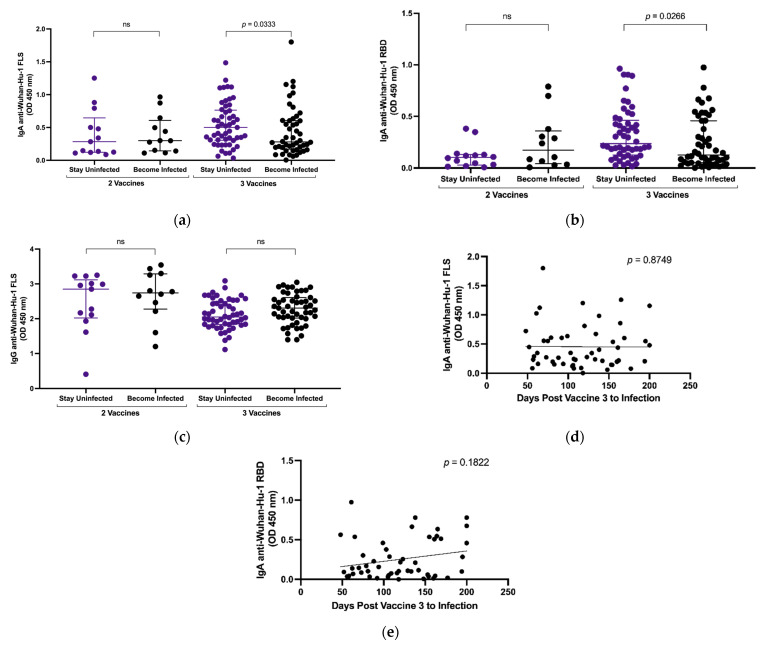
Comparison of plasma IgA and IgG anti-SARS-CoV-2 Wuhan-Hu-1 FLS and RBD in subjects that remained uninfected versus subjects experiencing breakthrough infection. Plasma IgA OD against Wuhan-Hu-1 FLS (**a**) and RBD (**b**) in subjects after 2 or 3 vaccines. Plasma IgG OD against Wuhan-Hu-1 FLS in subjects after 2 or 3 vaccines (**c**). Horizontal lines within the grouped data points represent median plus or minus interquartile range (IQR). Relationship between IgA levels against Wuhan-Hu-1 FLS (**d**), RBD (**e**) and days between vaccination and infection was assessed using Spearman correlation. Comparison of median values between groups was conducted using the Wilcoxon signed rank test. When significant, probability values for differences between groups are shown above lines spanning the groups compared.

**Figure 3 vaccines-11-01462-f003:**
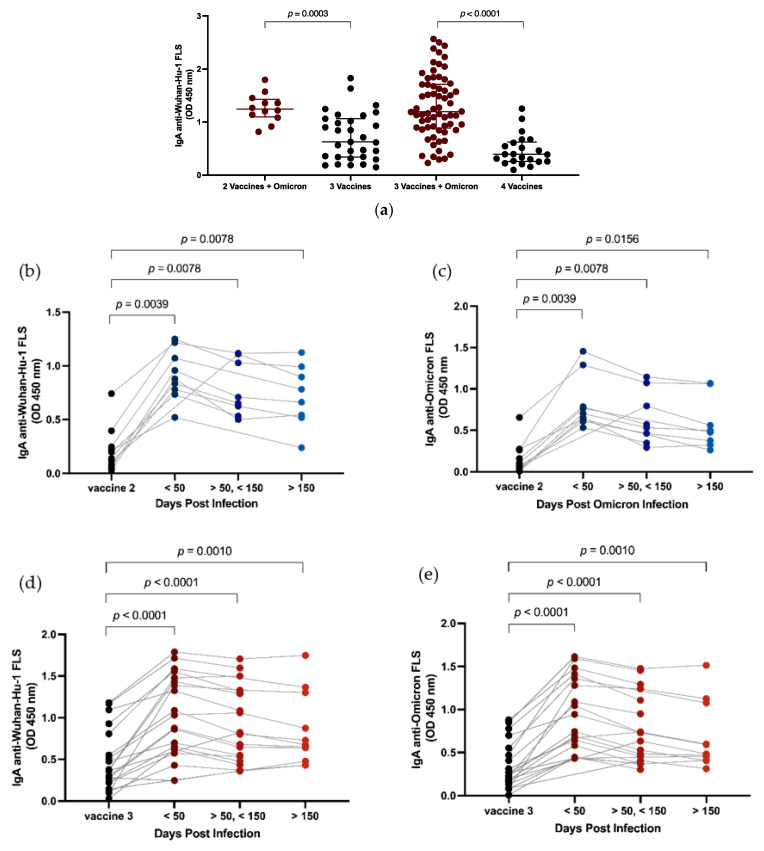
Effect of breakthrough infection on plasma IgA levels against SARS-CoV-2. Vaccination versus infection was compared in groups of subjects with 3 or 4 exposures to SARS-CoV-2 S (**a**). Longitudinal analysis of IgA anti-SARS-CoV-2 Wuhan-Hu-1 (**b**,**d**) and Omicron FLS (**c**,**e**) in subjects after 2 vaccines and Omicron infection (**b**,**c**) and 3 vaccines followed by Omicron infection (**d**,**e**). Horizontal lines within the grouped data points (**a**) represent median plus or minus interquartile range (IQR). Comparison of median values between groups was conducted using Mann–Whitney test (**a**) and Wilcoxon signed rank test (**b**–**e**). When significant, probability values for differences between groups are shown above lines spanning the groups compared.

**Figure 4 vaccines-11-01462-f004:**
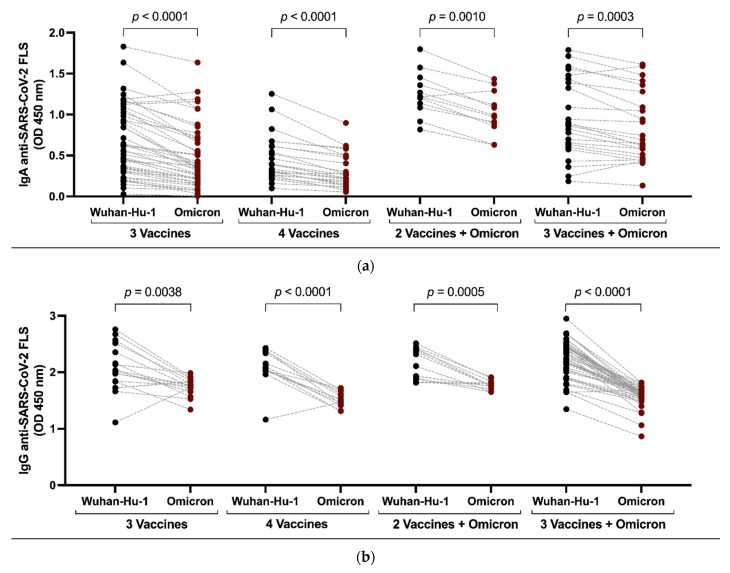
Comparison of plasma IgA and IgG Ab responses against SARS-CoV-2 Wuhan-Hu-1 versus Omicron BA.1 FLS after vaccination and after breakthrough infection. IgA (**a**) and IgG (**b**) responses against SARS-CoV-2 Wuhan-Hu-1 and Omicron BA.1 FLS were compared in vaccine recipients before and after Omicron breakthrough infection. Comparison of median values between groups was conducted using Wilcoxon signed rank test with values for the probability of significant differences between groups shown above lines spanning the groups compared.

**Table 1 vaccines-11-01462-t001:** General demographics and vaccine types received by subjects grouped by number of vaccines and incidence of SARS-CoV-2 breakthrough infection.

	2 Vaccines	2 Vaccines + Omicron	3 Vaccines	3 Vaccines + Omicron	4 Vaccines
n	13	12	49	47	22
Sex (M/F)	6/7	6/6	20/29	16/31	10/13
Age in years (mean, range)	41 (16–65)	40 (15–65)	55 (25–72)	53 (22–75)	63 (37–74)
^a^ Days Post Vaccine (mean)	61	79	57	66	59
^b^ Vaccine Types (mRNA-1273/BNT162b2/ChAdOx1)	1/10/2 2/11/0	2/11/0 3/10/0	5/38/6 13/36/0 22/27/0	0/39/8 4/41/2 21/26/0	2/18/2 8/14/0 9/13/0 9/13/0
^c^ Days Post Infection (mean)	-	40	-	52	-
Number of Immunogenic Exposures	2	3	3	4	4

^a^ Number of days between last vaccination and sample collection. ^b^ Number of people in each subgroup who received each type of licensed COVID-19 vaccine (mRNA-1273/BNT162b2/ChAdOx1) for vaccinations 1 through 4. ^c^ Number of days between documentation of infection and sample collection.

## Data Availability

Data supporting the findings of this study, preserving the anonymity of study participants, are available from the corresponding author through electronic correspondence for legitimate scientific purposes.

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
