# Peer review of "Characteristics of Vaccine- and Infection-Induced Systemic IgA Anti-SARS-CoV-2 Spike Responses"

_vaccines, 2023, doi:10.3390/vaccines11091462_

Round 1

Reviewer 1 Report

The authors produce interesting article about the differences in IgG and IgA responses after vaccinations and infections. For groups that work with vaccination in public health is a good work to fallow the status of vaccination and outbreaks by variants.

Considerations:
The lines 15-19 of the abstract are confuse, i think that can be clarify.
Line 178-Delete the extra dot.
In the conclusions the Original antigenic sin or antigenic imprinting should be mentionated in the considerations.

For my, the authors sound very clear in the text.

Author Response

The authors produce interesting article about the differences in IgG and IgA responses after vaccinations and infections. For groups that work with vaccination in public health is a good work to fallow the status of vaccination and outbreaks by variants.

Thank you for your positive comments following review of the manuscript and for your suggestions for improvement.

Considerations:
The lines 15-19 of the abstract are confuse, i think that can be clarify.

Thank you for the comment. We have made changes to the abstract (highlighted in yellow) to clarify the section in question.

Line 178-Delete the extra dot.

The extra dot has been deleted.

In the conclusions the Original antigenic sin or antigenic imprinting should be mentionated in the considerations.

In both the abstract conclusion and conclusion following the discussion, we mention that IgA imprinting occurred from vaccination.

Reviewer 2 Report

In this study, the authors measured circulating IgA against ancestral (Wuhan-Hu-1) and Omicron (BA.1) S proteins in 143 subjects with different vaccination and infection histories.

They report that intramuscular mRNA vaccination induces circulating IgA against SARS-CoV-2 S with higher levels associated with protection from breakthrough infection. Breakthrough infection generates stronger and more durable circulating IgA responses than vaccination alone. Intramuscular mRNA vaccination enacts imprinting within circulating IgA responses.

This original article is well written, interesting and pleasant to read but several points have to be corrected to improve the manuscript.

Abstract

Several sentences give the same message. Maybe the abstract can be divided in different paragraphs: Background/Results/Conclusion...Or add “In conclusion,” line 23

“and higher levels of circulating vaccine-induced IgA anti-SARS-CoV-2 S (p = 0.0333) or anti-receptor binding domain (RBD) (p = 0.0266) levels were associated with lesser risk for Omicron breakthrough infection”

Same message that

“Intramuscular mRNA vaccination induces circulating IgA against SARS-CoV-2 S with higher levels associated with protection from breakthrough infection.”

“Circulating IgA against SARS-CoV-2 S following breakthrough infection remained higher than vaccine-induced levels for over 150 days.

Same message that

“Breakthrough infection generates stronger and more durable circulating IgA responses than vaccination alone.” 

“Breakthrough infection following two vaccinations generated stronger IgA anti-SARS-CoV-2 S responses (p = 0.0002) than third vaccinations but did not selectively increase circulating IgA against Omicron over ancestral S, indicating immune imprinting of circulating IgA responses.”

Same message that

“We also found that intramuscular mRNA vaccination enacts imprinting within circulating IgA responses.”

Material & method

"Prior infection was confirmed by detection of antibodies selective for SARS-CoV-2 nucleocapsid (N)" Are recipients tested for pre-infection prior to vaccination? Because it is likely that on the date of the study, the majority of people have already been in contact with the original strain of the virus, this can therefore explain the great variability in the Ab levels (Sano K, Nat Commun. 2022) and create a bias in the interpretation of the data. Figure 1a shows Ab levels of pre-pandemic controls, but the baseline for each recipient is the level of Ab at the day of vaccination.

Regarding the method, the ELISAs used are not commercial kits but home-made assays without data on intra- and inter-plate reproducibility (between experiments). There are no calibrators (i.e., serum with known concentrations of IgA/IgG or monoclonal IgA/IgG) and Ab levels are measured by optical density. There are no internal controls described (same samples in plates for all experiments) to standardize results between different experiments. This lack of rigor may be responsible for a methodological bias

Table 1: group with 2 vaccines n = 13 but I count 13 dots in figure 2a, 14 dots in figure 2b and 12 dots in figure 2c

In the legend, the note “c” was indicated as "a"

Does # mean “number” of immunogenic exposures?

Types of vaccines: In the group with 3 vaccines, the number of individuals vaccinated with moderna (mRNA-1273) is higher compared to the other groups, especially for the first and second doses. The moderna vaccine is known to be more immunogenic than the pfizer vaccine (Canetti M, Nat Commun. 2022 / Steensels D,. JAMA. 2021) and this may explain why this group has higher levels of IgA and better protection against breakthrough infection.

Line 183: correct “samples have an average...”

Line 185: (Figure 2a,b) should be placed after IgA and (Figure 2c) after IgG

Line 188: the number of subjects is not indicated

Figure 2: the numbers of subjects in the groups with 2 vaccines are very low (n=12/14) compared to the groups with 3 vaccines, so it is difficult to conclude on the statistical analysis of the difference in levels of Ab.

Figure 2c: it is difficult to understand why the IgG levels are higher in the groups with 2 vaccines compared to the groups with 3 doses.

Figure 3: the numbers of subjects do not correspond to what is indicated in table 1 (in the group with 2 vaccines + Omicron, I see 13 points while there are only 12 subjects in table 1, in the group with 4 vaccines the number of points is greater than 22 subjects…

Figure 4, "b" in front of the second panel is missing. In the legend, line 260: correct “was made by Wilcoxon signed grading tests”

Line 340: correct “aanti-SARS-CoV-2”

The conclusion of the discussion correctly exposes the uncertain relationship between circulating and mucosal vaccine-induced IgA responses to clearly indicate that serum IgA is relevant in protection against a new SARS-CoV-2 infection. There is no very innovative message except perhaps the immune imprinting of IgA.

Finally, with the potential biases included in this study and the lack of data on mucosal immunity, it is difficult to conclude anything.

Author Response

In this study, the authors measured circulating IgA against ancestral (Wuhan-Hu-1) and Omicron (BA.1) S proteins in 143 subjects with different vaccination and infection histories.

They report that intramuscular mRNA vaccination induces circulating IgA against SARS-CoV-2 S with higher levels associated with protection from breakthrough infection. Breakthrough infection generates stronger and more durable circulating IgA responses than vaccination alone. Intramuscular mRNA vaccination enacts imprinting within circulating IgA responses.

This original article is well written, interesting and pleasant to read but several points have to be corrected to improve the manuscript.

Thank you for these positive comments and for your comprehensive review of the manuscript.  Addressing the points raised has improved the quality.

Abstract 

Several sentences give the same message. Maybe the abstract can be divided in different paragraphs: Background/Results/Conclusion...Or add “In conclusion,” line 23

 “and higher levels of circulating vaccine-induced IgA anti-SARS-CoV-2 S (p = 0.0333) or anti-receptor binding domain (RBD) (p = 0.0266) levels were associated with lesser risk for Omicron breakthrough infection”

Same message that

“Intramuscular mRNA vaccination induces circulating IgA against SARS-CoV-2 S with higher levels associated with protection from breakthrough infection.” 

“Circulating IgA against SARS-CoV-2 S following breakthrough infection remained higher than vaccine-induced levels for over 150 days.

Same message that

“Breakthrough infection generates stronger and more durable circulating IgA responses than vaccination alone.”  

“Breakthrough infection following two vaccinations generated stronger IgA anti-SARS-CoV-2 S responses (p = 0.0002) than third vaccinations but did not selectively increase circulating IgA against Omicron over ancestral S, indicating immune imprinting of circulating IgA responses.”

Same message that

“We also found that intramuscular mRNA vaccination enacts imprinting within circulating IgA responses.”

Together with a number of changes to the abstract (highlighted in yellow), we added “In conclusion” at the place indicated (now line 25).

Material & method 

"Prior infection was confirmed by detection of antibodies selective for SARS-CoV-2 nucleocapsid (N)" Are recipients tested for pre-infection prior to vaccination? Because it is likely that on the date of the study, the majority of people have already been in contact with the original strain of the virus, this can therefore explain the great variability in the Ab levels (Sano K, Nat Commun. 2022) and create a bias in the interpretation of the data. Figure 1a shows Ab levels of pre-pandemic controls, but the baseline for each recipient is the level of Ab at the day of vaccination.

We measured anti-SARS-CoV-2 N antibody levels longitudinally and specifically in relation to the N proteins of common b-coronaviruses to determine whether and when subjects were exposed to the virus (either the original Wuhan-Hu-1 strain or Omicron) as opposed to being vaccinated with a vaccine in which only the S protein of SARS-CoV-2 was represented.  This assay was used to confirm there was no SARS-CoV-2 infection prior to vaccination and then to confirm breakthrough infection following vaccination in those subjects reporting PCR or rapid antigen test documented infection.

Regarding the method, the ELISAs used are not commercial kits but home-made assays without data on intra- and inter-plate reproducibility (between experiments). There are no calibrators (i.e., serum with known concentrations of IgA/IgG or monoclonal IgA/IgG) and Ab levels are measured by optical density. There are no internal controls described (same samples in plates for all experiments) to standardize results between different experiments. This lack of rigor may be responsible for a methodological bias.

The reviewer notes that we use an ELISA test developed in-house without providing data on interassay variability.  While commercial kits may have the advantage of standardization, they lack flexibility eg. to substitute different antigens and control for amounts coated.  We have used the same in-house assay previously and provided data on interassay variability [12,15,17] indicating its reliability and reproducibility. For this manuscript, all between group comparisons shown were run in the same assay. (line 125-126).  

Table 1: group with 2 vaccines n = 13 but I count 13 dots in figure 2a, 14 dots in figure 2b and 12 dots in figure 2c 

Figures have been corrected and updated.

In the legend, the note “c” was indicated as "a" 

The legend has been corrected and updated.

Does # mean “number” of immunogenic exposures? 

The “#” did mean number.  The word “Number” has now been written out for clarity.

Types of vaccines: In the group with 3 vaccines, the number of individuals vaccinated with moderna (mRNA-1273) is higher compared to the other groups, especially for the first and second doses. The moderna vaccine is known to be more immunogenic than the pfizer vaccine (Canetti M, Nat Commun. 2022 / Steensels D,. JAMA. 2021) and this may explain why this group has higher levels of IgA and better protection against breakthrough infection. 

The reviewer makes the interesting observation here that vaccination with the Moderna mRNA vaccine is more common in some groups than others and that there is literature suggesting greater immunogenicity for this vaccine. This might have an effect on our analysis of the protective effect of IgA levels observed after 3 vaccines and not 2 vaccines, compounded by the lower number of subjects receiving only 2 vaccines and either remaining uninfected or experiencing infection.  Our analysis was only based on circulating IgA levels with the vaccines administered recorded for information purposes. In previous studies, we showed a slight increase in immunogenicity of the Moderna vaccine for IgG with larger comparison groups, but we have not addressed a comparison of IgA levels. This might be feasible with additional samples from within and outside this group, but was not part of this study. The main point remains that higher levels of vaccine induced IgA were associated with protection from breakthrough infection, irrespective of vaccine type.  

Line 183: correct “samples have an average...”

Corrected as shown with highlighting, now line 195, thank you.

Line 185: (Figure 2a,b) should be placed after IgA and (Figure 2c) after IgG 

Noted and changes highlighted, now line 197.

Line 188: the number of subjects is not indicated 

Number of subjects added and changes highlighted, lines 199 and 201.

Figure 2: the numbers of subjects in the groups with 2 vaccines are very low (n=12/14) compared to the groups with 3 vaccines, so it is difficult to conclude on the statistical analysis of the difference in levels of Ab. 

If there were more subjects receiving only 2 vaccines, the statistical comparison of Ab levels in those experiencing breakthrough infection and those not would be more robust, but the vast majority of participants in our study opted for boosting with a 3rd vaccination.

Figure 2c: it is difficult to understand why the IgG levels are higher in the groups with 2 vaccines compared to the groups with 3 doses.

The reviewer makes another interesting observation.  As mentioned, the groups with only 2 vaccines were smaller and we now notice they were also considerably younger.  In addition, the time between 2nd and 3rd  vaccinations was generally much longer than between the 2nd vaccination and sampling.

Figure 3: the numbers of subjects do not correspond to what is indicated in table 1 (in the group with 2 vaccines + Omicron, I see 13 points while there are only 12 subjects in table 1, in the group with 4 vaccines the number of points is greater than 22 subjects… 

Figures have been corrected and updated.

Figure 4, "b" in front of the second panel is missing. In the legend, line 260: correct “was made by Wilcoxon signed grading tests” 

Corrections noted and made.

Line 340: correct “aanti-SARS-CoV-2” 

The extra ”a” has been deleted.

The conclusion of the discussion correctly exposes the uncertain relationship between circulating and mucosal vaccine-induced IgA responses to clearly indicate that serum IgA is relevant in protection against a new SARS-CoV-2 infection. There is no very innovative message except perhaps the immune imprinting of IgA.

Finally, with the potential biases included in this study and the lack of data on mucosal immunity, it is difficult to conclude anything.

We agree that data on the relationship between mucosal and systemic immunity needs to be addressed in future studies and feel that our corroboration of the potential relevance of circulating IgA levels to protection from infection, illustration of the more robust nature of circulating IgA responses from infection than vaccination and demonstration of imprinting in circulating IgA responses is of interest to investigators working in this area.